# Proteomic Analysis of Female Synovial Fluid to Identify Novel Biomarkers for Osteoarthritis

**DOI:** 10.3390/life13030605

**Published:** 2023-02-22

**Authors:** P. Robinson Muller, Tae Jin Lee, Wenbo Zhi, Sandeep Kumar, Sagar Vyavahare, Ashok Sharma, Vikas Kumar, Carlos M. Isales, Monte Hunter, Sadanand Fulzele

**Affiliations:** 1Department of Medicine, Augusta University, Augusta, GA 30912, USA; 2Center for Biotechnology and Genomic Medicine, Augusta University, Augusta, GA 30912, USA; 3Anesthesiology & Perioperative Medicine, Augusta University, Augusta, GA 30912, USA; 4Center for Healthy Aging, Augusta University, Augusta, GA 30912, USA; 5Department of Orthopaedic Surgery, Augusta University, Augusta, GA 30912, USA; 6Cell Biology and Anatomy, Augusta University, Augusta, GA 30912, USA

**Keywords:** proteomic, female, synovial fluid, osteoarthritis

## Abstract

Osteoarthritis (OA) is a highly prevalent degenerative joint condition that disproportionately affects females. The pathophysiology of the disease is not well understood, which makes diagnosis and treatment difficult. Given the physical connection of synovial fluid (SF) with articular tissues, the SF’s composition can reflect relevant biological modifications, and has therefore been a focus of research. Previously, we demonstrated that extracellular vesicles isolated from the synovial fluid of OA patients carry different cargo (protein and miRNA) in a sex-specific manner. Given the increased prevalence and severity of OA in females, this study aims to identify differential protein content within the synovial fluid of female OA and non-osteoarthritic (non-OA) patients. We found that several proteins were differentially expressed in osteoarthritic females compared with age-matched controls. Presenilin, Coagulation Factor X, Lysine-Specific Demethylase 2B, Tenascin C, Leucine-Rich Repeat-Containing Protein 17 fragments, and T-Complex Protein 1 were negatively regulated in the OA group, with PGD Synthase, Tubulointerstitial Nephritis Antigen, and Nuclear Receptor Binding SET Domain Protein 1 positively regulated in the OA group. Database for Annotation, Visualization, and Integrated Discovery (DAVID) and QuickGO analyses established these proteins as significantly involved in many biological, cellular, and molecular processes. In conclusion, the protein content of female synovial fluid is altered in OA patients, which is likely to provide insights into gender-specific pathophysiology.

## 1. Introduction

Osteoarthritis (OA) is a highly prevalent degenerative condition affecting synovial joints [1]. The disease is characterized by the loss of cartilage, thickening of the synovial capsule, and the presence of hypertrophic bone and subchondral bone sclerosis [2]. The knee is the most commonly affected joint, followed by the hand and the hip [3]. According to the National Health Interview Survey results, the number of individuals in the US with OA symptoms specifically affecting the knee joint is approximately 14 million [4]. OA tends to target certain populations, with the elderly comprising the largest age demographic of those with knee OA [5] and women more likely than men to develop the condition [6]. Women and men with knee OA present differently, as women report more severe pain and higher rates of disability [7]. Women have been found to demonstrate unique patterns of cartilage degradation compared with men [8]. 

OA of any joint places a significant burden on people and their quality of life. Individuals with OA have been reported to experience twice as much difficulty while walking compared with those without OA. Furthermore, the affected population is restricted by limitations in activities, such as moving objects or getting dressed [9]. In addition to the constraints placed on individuals by OA, the impact has been felt on a societal level. Out of all diseases associated with disability, OA is growing the 3rd fastest, trailing only diabetes and dementia [10]. Additionally, the economic burden of OA is vast, with approximately $27 billion in health care expenditures ascribed to knee OA alone every year [11]. There are also further expenses, with a 2007 study attributing three days of work absence each year to all forms of OA, indirectly accruing a cost to society of USD 11.6 billion per year [12]. 

Even with this enormous problem plaguing people worldwide, the root cause of OA is still not well understood. The existing knowledge gaps pertaining to the disease have made diagnosis and treatment difficult. Although there is no absolute cure, current first-line OA treatments consist of exercise, weight loss, education, non-steroidal anti-inflammatory drugs (NSAIDs), and pain management [3]. Refractory cases can be managed with steroid injections and Total Joint Arthroplasties (TJAs) [13]. Even though TJAs impart substantial benefit to patients overall, a significant group unpredictably experience inferior outcomes [14]. The effectiveness of therapeutic intervention, in general, is greatly dependent on stage. The diagnosis of OA is usually only made when patients present with pain, and when joint damage is identified on radiographs. At this point, the damage is irreversible, and the focus is on palliation and stopping the progression of the disease. Much of the current research is focused on identifying the disease at an earlier stage, when the pathological process can be potentially stopped or slowed down.

A novel approach that is being considered is analyzing the metabolic changes in arthritic tissues and synovial fluid to identify novel biomarkers. Within the joint, synovial fluid (SF) is a viscous substance lubricating and cushioning the cartilage-lined cavity. Because of its physical connection with the articular tissues, the fluid’s composition can reflect relevant biologic alterations. Recently, we found that extracellular vesicles isolated from the synovial fluid of OA patients were differentially expressed in a sex-specific fashion [15,16]. Given the increased prevalence and severity of OA in females, we aimed to identify differential protein expression within the synovial fluid of female osteoarthritic (OA) and non-osteoarthritic (non-OA) patients. We collected age-matched synovial fluid of the two groups and analyzed them using mass spectrometry. We found several proteins were differentially expressed in the synovial fluid between the OA and non-OA female groups. Bioinformatic analyses revealed that these proteins as being involved in processes such as cell adhesion, cell differentiation, and locomotion, as well as playing a role in inflammation, tumorigenesis, and bone/cartilage formation.

## 2. Materials and Methods

### 2.1. Patient Samples

All methods were approved by the ethical committee (Code: 657441-24), in accordance with the guidelines and regulations of Augusta University. The Augusta University Institutional Review Board (IRB) approved the studies. Synovial fluid waste samples from age-matched female human knee joints [non-OA (*n* = 13), mean age 50.84 (±10.0) and OA (*n* = 15) mean age 52.8 (±5.6) patients] were obtained and deidentified, and did not necessitate informed consent approval. The SF samples were acquired from women undergoing total knee arthroplasties or arthrocentesis. These samples were obtained from osteoarthritic and non-osteoarthritic knee joints, excluding those with severe comorbidities (e.g., diabetes, hypertension, HIV, etc.) or blood contamination. We used synovial fluid, which was clear/colorless or faintly yellow colored. We avoided synovial fluid contaminated with blood. We did not search for additional characteristics of synovial fluid, such as white blood cell count or calcium pyrophosphate crystals. The samples were transported to the laboratory after synovial fluid collection in the operating room. Synovial fluid was diluted (1:1) with phosphate-buffered saline (PBS) and centrifugation at 3000 rpm for 20 min to exclude particles and cell debris. The resultant supernatant was analyzed with mass spectrometry.

### 2.2. Protein Extraction, Digestion, and Liquid Chromatography with Tandem Mass Spectrometry (LC-MS-MS) LC-MS/MS Analysis

Protein digestion and mass spectrometry were performed as per published methods [16]. Briefly, lyophilized samples were reconstituted with 100 µL of 50 mM ammonium bicarbonate buffer with 0.1% (*w/v*) RapiGest SF Surfactant (Waters) and 10 mM dithiothreitol; the cysteines were reduced at 60 °C for 30 min. The samples were then alkylated by iodoacetamide for 30 min and digested using trypsin (Thermo Scientific #90057) for 16 h at 37 °C. Trifluoroacetic acid was then added into the sample tube to lower the pH to under 2. The samples were subsequently incubated at 37 °C for 40 min to cleavage the detergent, followed by centrifuging at 15,000× *g* for 5 min. The supernatants were then collected into sample vials for LC-MS analysis. The digested peptide samples were analyzed on an Orbitrap Fusion tribrid mass spectrometer (Thermo Scientific, New York, NY, USA), to which an Ultimate 3000 nano-UPLC system (Thermo Scientific) was connected. Next, 2 µL of peptide samples was injected and trapped on a Pepmap100 C18 peptide trap (5 µm, 0.3 × 5 mm) and washed for 10 min at 20 µL/min using 2% acetonitrile with 0.1% formic acid. The peptides were then eluted from the trap and further resolved on a Pepman 100 RSLC C18 column (2.0 µm, 75 µm × 150 mm) at 40 °C. A gradient of between 2% and 40% acetonitrile with 0.1% formic acid was used over 120 min at a flow rate of 300 nL/min to separate the peptides. LC-MS/MS analysis was carried out by data-dependent acquisition (DDA) in positive mode, with the Orbitrap MS analyzer for precursor scans at 120,000 FWHM (full width at half maximum) from 300 to 1500 m/z, and the ion-trap MS analyzer for MS/MS scans at top-speed mode (3 s cycle time). Fragment of the precursor peptides was carried out using the collision-induced dissociation method, with a normalized energy level of 30%. Raw MS and MS/MS spectrum for individual samples were then processed with the Proteome Discoverer software by Thermo Scientific (v1.4), and subsequently submitted to the SequestHT search algorithm against the Uniprot human database (precursor ion mass tolerance: 10 ppm, product ion mass tolerance: 0.6 Da, static carbamidomethylation of cysteine (+57.021 Da), and dynamic oxidation of methionine (+15.995 Da). Peptide spectrum matching validation and false discovery rate estimation were performed using the built-in Percolator PSM validator algorithm.

### 2.3. Statistical and Bioinformatics Analyses

Spectral counting quantification was performed to compare protein abundance in different samples. Each protein in a specific sample had the peptide spectrum match (PSM) count normalized. This was done using the PSM count sums for that specific sample to compensate for possible variations during the LC-MS analysis. Further, quantile normalization was performed using the preprocessCore R package, and the differences in protein expression between the two groups (OA and non-OA) were analyzed using the LIMMA R package. Proteins that were upregulated or downregulated using a *p*-value cutoff of 0.05 were classified as differentially expressed for further analysis. Gene ontology pathway analyses for differentially expressed proteins and genes were conducted using the Database for Annotation, Visualization, and Integrated Discovery (DAVID) and QuickGO. Uniprot Knowledgebase (UniProtKB) protein descriptions and gene products were introduced into DAVID and QuickGO for statistical analysis and GO term annotation based on integrated cellular, molecular, and biological pathways of the differentially expressed proteins.

## 3. Results

### 3.1. Synovial Fluid Protein Content in Women Differs Significantly from OA Patients

In previous studies, we demonstrated that extracellular vesicle protein and exosomal miRNA content are altered in OA patients in a sex-specific manner [15,16]. In the present study, we completed mass spectrometry profiling and analysis of proteins obtained from the synovial fluid of female non-OA (*n* = 13) and OA (*n* = 15) patients. The synovial fluid was acquired from the knee joints of women undergoing either arthrocentesis or total knee arthroplasty procedures. The mass spectrometry data indicated several proteins expressed differentially in the two patient populations. The principle component analysis (PCA) and heat map of female OA were clustered distinctly from the non-OA (control) group (Figure 1). We found a significant disparity in 29 proteins, with 26 downregulated and 3 upregulated in the OA group (Table 1). For example, Presenilin (*p* = 0.02), Coagulation Factor X (*p* = 0.02), Lysine-Specific Demethylase 2B (*p* = 0.02), Tenascin C (*p* = 0.03), Leucine-Rich Repeat-Containing Protein 17 fragments (*p* = 0.02), and T-Complex Protein 1 (*p* = 0.04) were negatively regulated in the OA group. PGD Synthase (*p* = 0.03), Tubulointerstitial Nephritis Antigen (*p* = 0.04), and Nuclear Receptor Binding SET Domain Protein 1 (*p* = 0.03) positively regulated in the OA group (Figure 2). These results indicate that protein expression within the synovial fluid of women is significantly affected by the osteoarthritis condition. 

### 3.2. DAVID and QuickGO Analysis of Differentially Expressed Proteins

Database for Annotation, Visualization, and Integrated Discovery (DAVID) and Quick GO annotation analyses were completed to evaluate the regulatory functions of the differentially expressed proteins and their roles in biological, cellular, and molecular pathways. The analyses identified that these proteins are involved in processes such as cytoskeletal organization, molecule adhesion, nucleic acid binding, cell differentiation, and response to stress and wounding (Table 2).

## 4. Discussion

OA does not affect all individuals uniformly, and females are disproportionately afflicted with high prevalence and morbidity compared with males. To understand the pathophysiology and to identify an early diagnosis of OA, we performed proteomic analysis on the synovial fluid of female OA patients. Synovial fluid (SF) is secreted from the tissues of the knee joint, and its composition can provide important information about the health of the articular joint. Several techniques, including ELISA, mass spectrometry, and 3D gel electrophoresis, were used to identify biomarkers in the synovial fluid. Although previous studies have analyzed protein content in osteoarthritic joints, they failed to do so in a sex-specific manner. Thus, these studies could not provide information about distinct pathophysiologic processes in males or females. In this study, we analyzed the protein content of synovial fluid using mass spectrometry in age-matched OA and non-OA females. 

Our study found several proteins differentially regulated in the OA group compared with age-matched controls. Some of these proteins are important in cartilage and bone biology. For example, in the present study, Presenilin (PSEN) and Tenascin C were downregulated in the OA group’s synovial fluid. Presenilin is a transmembrane protein that plays a vital role in regulating the cleavage of several proteins in age-related diseases. It has been previously demonstrated that presenilin-deficient mice display an osteoporotic phenotype [17] with elevated osteoblast-dependent osteoclastic activity [18]. In another study comparing female congenic mice, PSEN was determined as a candidate gene that regulates trabecular thickness in a gender-specific fashion [18]. Taken altogether, these findings indicate that a decrease in Presenilin levels might be involved in bone and cartilage metabolism in a female with OA. 

The glycoprotein, Tenascin C (TNC), is expressed in the extracellular matrix of several tissues during the development and progression of diseases. TNC has been revealed in past studies to be positively regulated in the synovial fluid of OA and RA patients, as well as in diseased synovium and cartilage [19]. Reports indicate that TNC levels fluctuate at different stages of disease progression and cartilage repair [19]. It has been established that TNC fragments containing certain domains endogenously induce articular cartilage catabolism and synovial inflammation, with the full-length protein protecting against OA progression and stimulating cartilage repair [19]. TNC levels have been found to surge in canine synovial fluid during the early phases of OA and subsequent cartilage repair, and decrease as the cartilage matures, thus serving as a helpful indicator of OA progression. Although TNC has previously been found in elevated amounts in knee OA synovial fluid samples, the negative regulation of TNC in our OA group may point to its involvement in female pathology, an area that has not previously been reported. 

Other differentially expressed proteins between the two study groups have been associated with OA pathophysiology, and their regulation within the synovial fluid may give further insight into their specific functions. In the OA group of patients, Tubulointerstitial Nephritis Antigen (TINAG) and PGD synthase (PGDS) were upregulated, while Lysine-Specific Demethylase 2B (KDM2B) was downregulated, with all three proteins previously associated with OA. The dysregulation of TINAG gene expression has previously been associated with hand OA [20]. Li et al. [21] report an alteration of KDM2B gene expression in the synovial membranes of OA patients, which has been identified as a key transcription factor in the pathophysiology of the disease. PGDS has also been linked to OA. In male mice, the gene deletion of PGDS worsened the progression of OA, including effects on subchondral bone, synovial inflammation, and cartilage damage [22]. PDGS-negative mice displayed joint space narrowing, meniscus mineralization, and osteophyte formation [23]. Subsequent overexpression of PGDS reversed these effects [22]. The elevation of PGDS may also be a physiological response to joint insult, as it catalyzes the production of PGD_2,_ which is a major player in resolving inflammation [24]. These three proteins may be involved in pathological synovial joints in female OA patients. 

Other proteins that we identified as differentially regulated in the SF of the OA and non-OA female patients have not been previously implicated in OA, but have been linked to other pathological bone-related conditions. For example, Leucine-rich repeat-containing protein 17 (LRRC17) fragments and T-Complex Protein 1 (TCP1) were both downregulated in the OA group, while nuclear receptor binding SET domain protein 1 (NSD1) was upregulated. LRRC17 prevents the activation of RANK-L-dependent osteoclasts, thus protecting against osteoporotic phenotypes. In fact, the risk of postmenopausal women experiencing osteoporotic fractures was found to be 46% higher in those with lower LRRC17 levels than those with higher levels [25]. The downregulation of the LRRC17 protein in OA synovial fluid may reflect an affected pathway that is shared by the pathology of osteoporosis. TCP1 is also involved in bone pathology, and specifically, Ankylosing Spondylitis (AS)-induced heterotopic ossification. The protein has been shown to be underexpressed in the bone marrow-derived mesenchymal stem cells of Ankylosing Spondylitis compared with controls [26]. NSD1 is yet another protein involved in bone pathology, but novel in the context of OA. A deficiency of this protein is the primary cause of Sotos syndrome, which is marked by advanced bone age [27]. The pathogenesis of this condition is unknown, but the potential metabolic role that NSD1 plays in both Sotos syndrome and OA should be studied. It has been reported that bone alterations are among the earliest changes seen in osteoarthritic joints, even occurring before cartilaginous damage [28]. The relationship between OA development and bone mineral density has been staunchly disputed, and study findings are often contradictory. This may be due to a variety of OA subtypes with separate mechanisms as well as bone mineral density (BMD) fluctuations in different joint locations and disease stages [29,30]. The complicated relationship between bone metabolism and the pathogenesis of OA should nevertheless be explored, beginning with the differential expression in female synovial fluid. 

Our study provides a strong foundation for future exploration of OA pathophysiology, especially in women. However, the study does have some limitations. First, the sample size was limited in the number of female synovial fluid samples analyzed, and a more extensive study would be an excellent next step. A study with SF samples from females with varying stages of OA would also be beneficial in identifying the correlation between disease severity and protein expression. Investigations with large groups of diverse patient populations of females are necessary as the disease is very heterogeneous. Our study primarily focused on identifying novel proteins in the fluid samples using a proteomic approach, but future studies are needed to identify their direct roles in the pathogenesis of articular cartilage. Additional functional studies using in vitro and in vivo models could be carried out to distinguish the functions of these differentially expressed proteins in OA. More insight on the biological effect of OA can also be gained by characterizing protein expression in other body fluids, such as urine or serum.

## Figures and Tables

**Figure 1 life-13-00605-f001:**
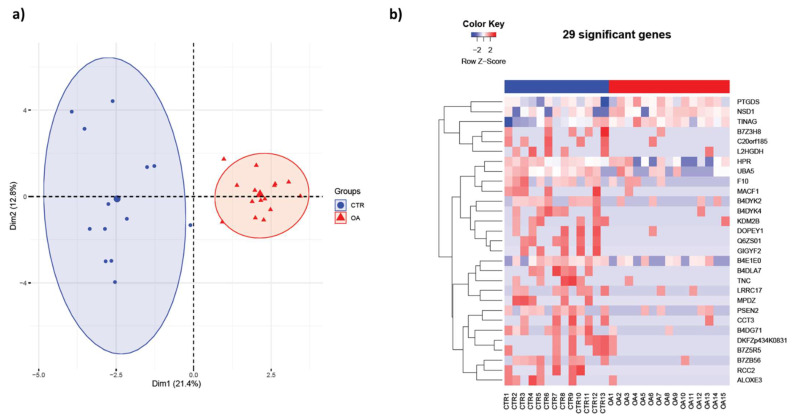
Synovial fluid protein content differs in female osteoarthritis (OA) patients. (**a**) Principal component analysis (PCA) mapping of female OA and non-OA (control) profiling. Female OA group (indicated by red triangles) was clustered distinctly from non-OA (control) group (indicated by blue circles). (**b**) Heatmap of (**a**) female OA (*n* = 15) and non-OA (*n* = 13) patients. Differences between normal and OA patients were examined using the empirical Bayes moderated test, and only those that were significantly different at the *p* value 0.05 level were selected.

**Figure 2 life-13-00605-f002:**
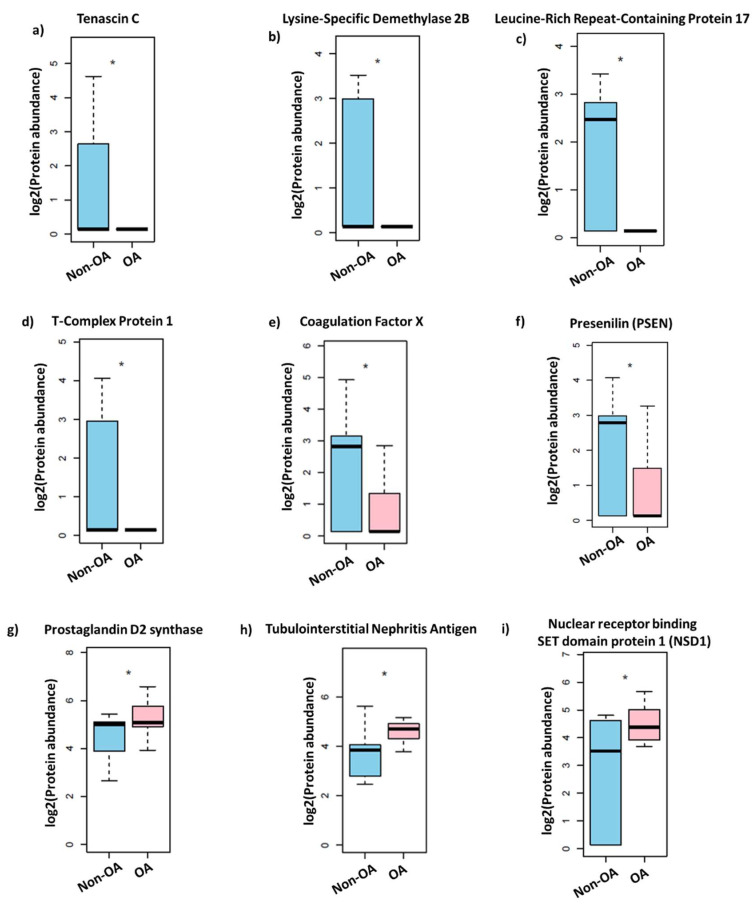
Differentially regulated proteins in the synovial fluid of female OA patients. (**a**) Tenascin C, (**b**) Lysine-Specific Demethylase 2B, (**c**) Leucine-Rich Repeat-Containing Protein 17, (**d**) T-Complex Protein 1, (**e**) Coagulation Factor X, (**f**) Presenilin, (**g**) Prostaglandin D2 synthase, (**h**), Tubulointerstitial Nephritis Antigen, and (**i**) Nuclear Receptor Binding SET domain protein 1. Differences between normal and OA patients were examined using the empirical Bayes moderated test [non-OA (*n* = 13) and OA (*n* = 15) patients, * *p* value = 0.04].

**Table 1 life-13-00605-t001:** List of differentially regulated proteins in the synovial fluid of female OA compared with non-OA patients.

Accession	Description	Fold Change	*p* Value
B2RWP5	Nuclear receptor binding SET domain protein 1 (NSD1)	2.65	0.028
A0A024R8G3	Prostaglandin D2 synthase 21kDa (Brain), isoform CRA_a	1.95	0.035
Q5T466	Tubulointerstitial Nephritis Antigen	1.90	0.040
J3KPH2	Arachidonate lipoxygenase 3, isoform CRA_a	0.53	0.030
Q14DE0	Chromosome 20 open reading frame 185	0.52	0.037
B7Z5R5	cDNA FLJ61652, highly similar to Mus musculus DEP domain containing 1a (Depdc1a), mRNA	0.51	0.027
C9JVN9	L-2-hydroxyglutarate dehydrogenase, mitochondrial	0.51	0.042
Q69YG3	Putative uncharacterized protein DKFZp434K0831 (Fragment)	0.51	0.027
B4DUR8	T-Complex Protein 1 subunit gamma	0.50	0.039
I1E4Y6	PERQ amino acid-rich with GYF domain-containing protein 2	0.49	0.006
H0UI11	Dopey family member 1, isoform CRA_a	0.48	0.033
A0A024R884	Tenascin C (Hexabrachion), isoform CRA_a	0.48	0.033
B4DYK4	cDNA FLJ56807, highly similar to Rab6-interacting protein 1	0.47	0.015
C9JT74	Leucine-rich repeat-containing protein 17 (Fragment)	0.47	0.020
H0YGQ3	Multiple PDZ domain protein (Fragment)	0.46	0.006
A0A0C4DGG3	Lysine-Specific Demethylase 2B	0.45	0.018
B7Z3H8	cDNA FLJ57997, highly similar to Transmembrane GTPase MFN2 (EC 3.6.5.-)	0.45	0.030
H3BPE1	Microtubule-actin cross-linking factor 1, isoforms 1/2/3/5	0.45	0.027
A5PLK7	RCC2 protein (Fragment)	0.44	0.006
B4DYK2	cDNA FLJ56270, highly similar to Homo sapiens pleckstrin homology domain containing, family G (with RhoGef domain) member 2, mRNA	0.43	0.028
B4E1E0	CTP synthase	0.42	0.038
Q6ZS01	cDNA FLJ45938 fis, similar to Mus musculus zinc finger protein 292 (Zfp292) (Fragment)	0.42	0.002
E5RFW4	Presenilin (Fragment)	0.42	0.021
Q5JVE7	Coagulation Factor X	0.40	0.025
B4DLA7	cDNA FLJ59923, highly similar to Cytohesin-1	0.39	0.002
E7EWE1	Ubiquitin-like modifier-activating enzyme 5	0.34	0.020
B7ZB56	cDNA, FLJ79420, highly similar to Homo sapiens nebulette (NEBL), transcript variant 2, mRNA	0.33	0.001
J3KTC3	Haptoglobin-related protein	0.32	0.025
B4DG71	cDNA FLJ56619, highly similar to cAMP-specific 3’,5’-cyclic phosphodiesterase4B (EC 3.1.4.17)	0.26	0.002

**Table 2 life-13-00605-t002:** Selected Database for Annotation, Visualization, and Integrated Discovery (DAVID) Gene Ontology (GO) pathways affected in the synovial fluid of female OA patients.

Gene Ontology	Term	Ont	*p* Value
GO:0006950	response to stress	BP	0.007218
GO:0009611	response to wounding	BP	0.007153
GO:0016043	cellular component organization	BP	0.024775
GO:0019538	protein metabolic process	BP	0.013128
GO:0030154	cell differentiation	BP	0.037616
GO:0042060	wound healing	BP	0.004072
GO:0048518	positive regulation of biological process	BP	0.00079
GO:0048523	negative regulation of cellular process	BP	0.023211
GO:0048869	cellular developmental process	BP	0.040836
GO:0050789	regulation of biological process	BP	0.012056
GO:0050794	regulation of cellular process	BP	0.005146
GO:0065007	biological regulation	BP	0.021133
GO:0071840	cellular component organization or biogenesis	BP	0.029521
GO:1901564	organonitrogen compound metabolic process	BP	0.033099
GO:0048522	positive regulation of cellular process	BP	0.000313
GO:0030334	regulation of cell migration	BP	0.022019
GO:0032879	regulation of localization	BP	0.025221
GO:0040012	regulation of locomotion	BP	0.028507
GO:0051270	regulation of cellular component movement	BP	0.031709
GO:2000145	regulation of cell motility	BP	0.02594
GO:0007155	cell adhesion	BP	0.010892
GO:0022610	biological adhesion	BP	0.011072
GO:0030155	regulation of cell adhesion	BP	0.009815
GO:0051128	regulation of cellular component organization	BP	0.012686
GO:0034330	cell junction organization	BP	0.008997
GO:0040007	growth	BP	0.002564
GO:0048589	developmental growth	BP	0.007582
GO:0051276	chromosome organization	BP	0.040125
GO:0012505	endomembrane system	CC	0.012368
GO:0031982	vesicle	CC	0.027865
GO:0043227	membrane-bounded organelle	CC	0.026887
GO:0015630	microtubule cytoskeleton	CC	0.006241
GO:0043228	non-membrane-bounded organelle	CC	0.043619
GO:0043232	intracellular non-membrane-bounded organelle	CC	0.043272
GO:0042995	cell projection	CC	0.049255
GO:0043005	neuron projection	CC	0.049972
GO:0005768	endosome	CC	0.020981
GO:0005794	Golgi apparatus	CC	0.002081
GO:0005783	endoplasmic reticulum	CC	0.009801
GO:0030425	dendrite	CC	0.005927
GO:0036477	somatodendritic compartment	CC	0.014925
GO:0044297	cell body	CC	0.005378
GO:0097447	dendritic tree	CC	0.005981
GO:0099080	supramolecular complex	CC	0.000852
GO:0099081	supramolecular polymer	CC	0.0027
GO:0099512	supramolecular fiber	CC	0.002622
GO:0099513	polymeric cytoskeletal fiber	CC	0.010599
GO:0005874	microtubule	CC	0.002197
GO:0005515	protein binding	MF	0.035746
GO:0003824	catalytic activity	MF	0.039472
GO:0003676	nucleic acid binding	MF	0.035847
GO:0003723	RNA binding	MF	0.005421
GO:0008270	zinc ion binding	MF	0.013125
GO:0046914	transition metal ion binding	MF	0.028036
GO:0140096	catalytic activity, acting on a protein	MF	0.008336
GO:0004175	endopeptidase activity	MF	0.00233
GO:0008233	peptidase activity	MF	0.006311

## Data Availability

The data that support the findings of this study are available from the corresponding author upon reasonable request.

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
