# Peer review of "Proteomic Analysis of Female Synovial Fluid to Identify Novel Biomarkers for Osteoarthritis"

_life, 2023, doi:10.3390/life13030605_

Round 1

Reviewer 1 Report

It’s not clear what was the diagnosis of the NON- OA patients.

Why did they have an effusion? Did they have an arthropathy, if so, which one? It is important to well define this control population.

How was OA patients diagnosed? (criteria)

The Authors should describe the characteristics of these synovial fluids including white blood cell count. Was some sample positive to calcium pyrophosphate crystals?

Figures: Student t-test is a parametric test. Authors have to report that the parameters considered were normally distributed. Otherwise a non-parametric test should be performed.

Patients’ age seems relatively low (mean around 50) and some of them were submitted to TKA. This implies that they have a severe OA? Why the kellgren-lawrence score is not reported?

Without a male control group, the Authors cannot state any sex-specific results. Discussion should be modified and protein changes cannot be referred only to pathological process in female OA patients.

Minor

Line 87: mean age I suppose

Line 89: SF samples rather than tissue samples

Line 196: Several modern techniques, ELISA and electrophoresis are not really modern

Line 283: remove conclusions

Reviewer 2 Report

Methods: going forward, please obtain informed consent for surgical specimens. 

Results: please cite line 143-144

how were subjects age matched? what demographic information was collected on subjects? it would be nice to see a table 1 demographics chart

Conclusion: please remove conclusion placeholder

Round 2

Reviewer 1 Report

The Authors should modify the text besides answering to the reviewer. OA patients and controls have to be clearly defined in patient samples section.

The diagnostic criteria for OA are missing (ACR? EULAR? others
?) 

Again: Without a male control group, the Authors cannot state any sex-specific results. SUMMARY "The protein content of female synovial fluid is altered in OA patients, likely giving insight into 28 the gender-specific pathophysiology" 

Author Response

Included as suggested.